# Children and Drug Trafficking in Brazil: Can International Humanitarian Law Provide Protections for Children Involved in Drug Trafficking?

**Veridiana Bessa Franciozo Diniz** [1] **and Jody Lynn McBrien** [2,*]

1   Interdisciplinary Social Sciences, University of South Florida, Sarasota, FL 34240, USA
2   School of Interdisciplinary Global Studies, University of South Florida, Sarasota, FL 34240, USA
*   Correspondence: jlmcbrie@usf.edu

**Abstract:** Brazil has seen a rise in children in narco-trafficking due to increased conflicts between factions and local law enforcement. Mainstream media and scholars tend to frame actions of these factions as organized crime, ignoring the generalized violence the communities and children experience. The aim of this study is to conduct a scoping review to consider whether or not Brazilian children involved in drug trafficking can be classified as child soldiers. Drawing from the international definition of Armed Conflict in Article 3 of the Geneva Convention of 1949 and Article 1 of the Additional Protocol II, and comparing situations of realities faced by Brazilian children involved in narco-trafficking, we argue that their reality is analogous to that of child soldiers, as defined by the Paris Principles on the Involvement of Children in Armed Conflict 2007; thus, going beyond the organized crime definition. In characterizing them as child soldiers, we argue for improving the children's ability to be reintegrated into society, with the collective help of the international community.

**Keywords:** child soldiers; drug trafficking; Brazil; humanitarian law

## 1. Introduction

Since 2016, Brazil has faced an increase in deaths. The year 2017 reached the historic height of violent deaths with 63,880 victims, equivalent to 175 deaths per day. The number represents an increase of 3.7% compared to the year of 2016, making Brazil the country with highest number of absolute death numbers. The rate of 30.8 deaths per 100 thousand inhabitants puts the country among the top 10 most violent countries in the world [1].

The high level of violence in the country is due to conflict between drug-trafficking factions. Since 2016, the two main factions, the Comando Vermelho (CV) and Primeiro Comando da Capital (PCC), have been disputing control of important drug routes in South America, including the Rota Solimões, between Colombia's, Peru's, and Brazil's frontiers [2]. The disputes between the two factions from the south-eastern region of the country led to their spread throughout Brazil, migrating to the north and northeast regions seeking new opportunities and markets [3]. Differently from the southern region in Brazil, where the presence of these factions in the states is stronger (especially in the case of PCC and its hegemony in São Paulo), the spread to the northern region was marked by a clash with already existing smaller factions [3]. Fifteen out of the 26 states in the country saw an increase in violent deaths with this event, with Rio Grande do Norte being the most violent one, at 62.8 homicides per 100 thousand habitants in 2016 [2].

The main contributor to the high number of violent deaths in Brazil is the increase in massacres by drug-running factions in penitentiary systems. In 2017, members of the Família do Norte (FDN) (ally to the CV) faction killed and mutilated 56 members of the PCC at the Anísio Jobim penitentiary, in Manaus, state of Amazônas; 33 were executed at the Agrícola de Monte Cristo penitentiary, in Boa Vista, state of Roraima; and another 26 at

Alcaçuz penitentiary, in Natal, state of Rio Grande do Norte [4]. The massacres follow the same dynamics in which members of one faction invade the sectors of the prison controlled by the other faction, killing the inmate members of the rival faction with cruelty [3].

Killing prisoner members of rival factions eliminates a large number of leaders, while also intimidating rivals. This creates a socialization process in which younger faction members seeking to expand their influence in the drug market are required to make their presence be noted by promoting more violent actions and advertising their actions through sharing videos of the cruelties committed [3].

The high level of violence in the country is more critical among the younger population. For people aged from 15 to 29 the death rate is of 69.9 per 100 thousand habitants, a record in the past 10 years. In 2017, 35,783 young people were murdered, an increase of 37.5%. Adolescents from the age of 15 to 19 comprise more than half of deaths, while those of 20 to 24 comprise 49.4% [2]. The number of adolescents doing socio-educative measures—in which young offenders are sentenced to complete pedagogical measures, either in open, semi-open or closed regimes of detention centres [5]—doubled to 192 thousand, in the majority of the cases due to involvement in drug-trafficking. Ninety percent of them are male.

Reducing this life-threatening situation in Brazil is complex. Three situations currently exist, two of which can decrease violence, as follows: (1) accept the conflict reality and allow multiple factions at war, as in the state of Rio de Janeiro; (2) allow the monopoly of one faction controlling the drug market, including the case of PCC in the state of Sao Paulo, which contributes to a relative peace with the lack of competition; or (3) multiple factions at peace, in the case of the state of Ceará. The problem of the last scenario is that the armistices between the factions are fragile, creating an unstable reality where a new conflict can emerge at any time [6]. Negotiating with faction leadership and promoting public measures that reduce illiteracy, flaws in the educational system, poor living conditions, lack of sanitation, and employment could also be implemented to avoid problems faced in the prison system [3].

This scoping review explored how children are inserted in this world of drug trafficking and ways in which their reality is parallel to those children categorized as child soldiers by international humanitarian law. To do so, it is necessary to explain how the drug trafficking context in Brazil can be understood as an armed conflict, since the legal definition of child soldiers, according to the Paris Principles on the Involvement of Children in Armed Conflicts [7], relies on their direct association to armed conflicts or groups. Therefore, this study is subdivided into two main parts: (1) answering how and why drug trafficking in Brazil should be considered a civil war (and thus an armed conflict) rather than organized crime; and (2) why children inserted into this context are child soldiers rather than juvenile delinquents. The study will then explore the roles children play in drug factions; how these are similar to the reality of children classified as child soldiers; and how the international community has positively contributed to revert the situations of child soldiers. While the root solution to this problem is far from the scope of this paper, protection should be guaranteed for the children.

## 2. Methodology

This study was conducted between Fall of 2020 and Spring of 2021, when the COVID-19 pandemic was at its peak and Brazil was on full lockdown. Due to this, no ethnographic first-hand search was conducted for this study. Instead, we chose to conduct a scoping review [8], as we found little evidence that other scholars have researched our questions, making this method appropriate for our question, given the clear gap in information [9]. For instance, Pedersen and Sommerfelt [10] stated, "In some countries, such as Brazil, children are recruited into armed groups that are not engaged in war, but rather in violent, organized crime" (p. 255). Similarly, Arias and Rodrigues [11] discuss child involvement as criminal activity, but not as child soldier recruits. We found this throughout our review of literature. In contrast, we contend that the situation of children drug traffickers in Brazil

needs to be viewed from the perspective of recruited child soldiers, as this perspective could offer them enhanced protection and rehabilitation of international treaties.

Thirty-nine documents are included in our review. We were only able to locate eight peer-reviewed sources that touched on our focus about children used in the drug wars in Brazil. Most of our research came from 12 historical articles and chapters about the drug trade in Brazil, 13 newspaper articles about issues that included the issues in Brazilian prisons, and 6 international treaties and policy documents.

We examined the situation of drug trafficking in Brazil through research articles as well as local and reputed newspapers from *G1, Estadão, UOL Noticias, Epoca*, BBC News, *Gazeta do Povo, Agencia* Brasil and *Folha de S. Paulo*. Because the first author is Brazilian, we were able to include documentation that has not been translated into English. National reports conducted by federal and local institutes were also used in this research. These were found in databases, specifically JSTOR, available through our university that provide records and documents in non-English languages. In order to provide context, academic articles and books with detailed historical background on how the drug factions were formed are included in this research. Finally, international reports, treaties and conventions were used to describe legal definitions, arguments to include Brazilian child drug traffickers under the umbrella of child soldiers, and analysis. We did not include literature that considered the children to be criminals or juvenile delinquents. There is considerable literature connecting the children to crime. However, we propose that they should be viewed instead as recruited child soldiers who meet a commonly accepted definition of the term "child soldier": "any child, boy or girl, under the age of 18, who is compulsorily, forcibly, voluntarily recruited or otherwise used in hostilities by armed forces, paramilitaries, civil defence units or other armed groups" (p. 5, [12]).

The time frame established and used for this research was five years, meaning that the materials used would not be older than five years from when the research was done. There are, however, two exceptions: (1) the national census in Brazil is conducted every other 10 years, being the last one from 2010; and (2) because there is a necessity of drawing extensive historical background, some older information about the factions were from previous years. Therefore, in some cases, the references used were older than the established time frame.

We used key words in combination, primarily "Brazil" and "drug-trafficking or drugs", and "children or youth or child". We selected articles using those key words that we found in JSTOR, Academic Search Premier, and the Web of Science as well as Google searches on Portuguese-language articles and international documents regarding child soldiers. Although we tried including the term "child soldier" with "Brazil", we did not find usable sources with that combination. For instance, using key terms "Brazil", "child soldiers", and "drugs", no results were found on the Academic Premier database. On JSTOR, we retrieved articles, but they distinguished between the situation of Brazilian child drug traffickers and child soldiers. As a result, we changed the key term "child soldiers" to "children" or "child" or "youth" in database and Google searches. We also added "United Nations" as well as "international policies" in combination with "child soldiers" to searches to find international treaties about child soldiers.

## 3. The Context of Drug Trafficking in Brazil

Criminal drug factions first appear in the context of Brazilian prisons. The Comando Vermelho (CV), considered the first Brazilian faction, was conceived due to exchange of knowledge between political prisoners and common criminals in a high security prison known as the "Devil's Caldron" [13]. In the context of the military dictatorship which lasted from 1964 to 1986, marked by a lack of basic rights, persecution, and massacre of those opposing the system, the CV revendicated better life conditions against the unfair treatment by the State, both inside and outside the prison system [13]. The faction later served as an example to other factions that appeared in the country in the following years,

following the same ideologies [14] and structural organization based on a set of laws, moral codes, and internal hierarchy [13].

Currently in Brazil there are six main factions with activities across the country. The Primeiro Comando Capital (PCC) is the most powerful and influential one in the world of crime, being present in 23 out of the 26 states and having control of three of them. Next is the original CV, present in seven states, followed by regional CVs in five states. The Família Monstro is present in two states, Okaida in two, and Família do Norte (FDN) in one state. However, there are another 27 active factions in the country, acting in specific states [15].

We chose to analyse the CV, due to its historical relevance as the first Brazilian faction and as an example to all the factions that later emerged; and the Primeiro Comando Capital (PCC), considered the most powerful and relevant faction in Brazil recently. The amount of information about the structural organization of PCC after an intensive investigation in 2013 [16] provides important information on how factions organize themselves.

### 3.1. The Comando Vermelho (CV)

Cândido Mendes prison was known for its inhumane treatment of prisoners, giving origin to its nickname "Caldeirão do Diabo" (Devil's caldron). Prisoners were kept in wood "shed-like" structures, with flooring made of sand, and barbwire as fences. The prison was built to accommodate 540 inmates; however, in 1979, there were 1284 imprisoned there. Beyond overcrowding, there was also a lack of meals, beddings, uniforms, toilet paper and blankets for the prisoners; and even the soldiers suffered from the State's abandonment [13,17].

The direct contact between prisoner groups led to a great exchange of knowledge. Political prisoners provided information about leftish ideologies, methodologies, and guerrilla organizations, while common prisoners provided knowledge about the world of crime [13]. The result was the emergence of a highly organized and structured network, different from common crimes that were previously committed such as car robbery, and including guerrilla style actions with detailed planning [13].

In 1979, a revolt among the prisoners in Cândido Mendes, ignited by the poor prison conditions, led to the emergence of Comando Vermelho (CV) [13]. Applying the same strategies as the ones used by Colombian prisoners, CV was able to obtain popular support by providing better living conditions to the community through the profits obtained by selling drugs [14]. Comando Vermelho became the primary distributor of cocaine to Europe in the 1980s. High profits enabled CV to purchase munitions, further distancing the State's interference in communities, making CV a "parallel State" [14]. Even though CV is still considered one of the main factions in Brazil, the creation of different police task forces to ensure peace in the dominated areas has been responsible for the weakening of the faction, and it is being overrun by the Primeiro Comando da Capital (PCC) [14].

### 3.2. The Primeiro Comando da Capital (PCC)

The Primeiro Comando Capital (PCC) was created in 1993 in the Casa Custódia de Taubaté, in Taubaté prison during a soccer match. Eight prisoners named their team Comando da Capital and had the goal to "fight the oppression within the Paulista's prison system" and avenge the death of 111 prisoners during the Carandiru Massacre [14]. The event happened in 1992 after a fight between prisoners in one of Casa de Detenção de São Paulo pavilions, which led to the invasion by the military police and death of 111 prisoners [18].

Social conveyance of its ideals against the poor treatment by police officers made it easier for prisoners to join the faction, resulting in its rapid growth [13]. The faction's main goal was to create a syndicate-like structure to defend the rights of prisoners against repression and inhumane treatment [14]. The failed attempt of officers to control prisoners was not enough to prevent the faction from attracting people (Maia, 2009): in 1997, the faction already had 8,000 members; in 2006, 120,000 in prisons [14]. It is estimated that the PCC has a profit of 300 million reais (about $60 million US).

### 3.3. The Origin of Faction Conflict

Faction conflict as a major event is dated to 2016, with the death of Jorge Rafaat Toumani, which ended the alliance between CV and PCC [19]. Originally there was a harmonious coexistence of the CV and PCC, in which both factions benefited from the relationship. The CV allowed PCC to sell drugs in some of the areas it dominated, while the PCC would negotiate weaponry for the CV. Through the trade of cocaine both factions would collectively participate in bombing and shooting public buildings in Rio de Janeiro and São Paulo [14].

The conflict between the two organizations is due to two different events. First was the assassination of Jorge Rafaat Toumani in 2016, who was considered the main drug trafficker in the frontier between Brazil and Paraguay, providing drugs to both factions. By killing Toumani, the factions intended to dominate the supply of drugs in the region without the need of an intermediator. However, after the murder, the PCC made the route exclusive to itself, betraying the agreement [14]. A second explanation for the conflict was the alliance of CV with other factions that are enemies to the PCC, trying to prevent the PCC from reaching a hegemonic status in the country [14].

The tension between the two factions can be seen in both the streets and within prisons. The bad treatment of prisoners with violations of basic rights fuelled the emergence of new factions and high adherence to the already existing ones. On the other hand, the State's inability to stop the growth of factions contributes to the uprising of factions as State-like actors [14].

The increasing competition and struggle over territorial control led to a growth in using children in factions, especially as combatants. Highly militarized organizations and an increase of conflicts, both with police and other factions, led to higher numbers of incarceration and deaths by gun violence. Therefore, children have been used to replace the positions once occupied by older members [20].

### 3.4. The State's Measures and War-like Conditions

The communities (favelas) in which the presence of factions is strong are marked by a historically unequal development, which some believe must be solved by the State. Since the 80's, with the rise of factions, cocaine consumption, and territory disputes, there has been a more prominent presence of civil and military police to prevent violence (de Araujo, 2016). At the same time, the 90's saw an increase of the rhetoric advocating for repressive measures against favelas as a measure to eliminate drug traffickers [21].

The highly militarized character of the illegal market results in a clash not only between factions but also between factions and police, specifically grave in the case of Rio de Janeiro where these encounters occur daily. The strong repression of drug trafficking in many communities has created a war-like environment in the favelas' territory. Two realities result from this violent "war on drugs." First is a high number of extrajudicial executions by the police during missions in favelas, justified as acts of resistance, rarely investigated. Second is an increase of faction members' populations in prisons, which contributes to the factions seeking younger men to substitute for those imprisoned, exposing younger populations to violence [21]. The violation of human rights, accompanied by the violence within the prison systems, contributes to the increased desire for revolt against the State [22]. The factions emerge as a way to control the prison system, showing themselves as strong, rich and protectors of their members, revendicating better conditions [17].

A high presence of factions remains in favelas, as law enforcement in favelas does not translate into an improvement of life conditions. The lack of access to basic rights, public health, employment, and basic sanitation, which marks the State's omission in these communities, contributes to the emergence of factions as substitutes to fulfil State obligations. Drug trafficking is how factions sustain their organizations, instead of charging the community [22]. Therefore, with this semi-dictatorial power factions hold in favelas, they can stimulate local economy, promote entertainment, and provide income, securing their legitimacy and social support [11].

## 4. Defining Armed Conflict

The definition of armed conflicts of a non-international character can be found in Article 3 of the Geneva Conventions of 1949 and Article 1 of the Additional Protocol II [23,24]. Article 3 applies to "armed conflict not of an international character occurring in the territory of one of the High Contracting Parties" [23] occurring between non-governmental groups; or between non-governmental and governmental armed forces, as long as they take place on the territory of one of the parties to the Convention [23].

Article 1 of the Additional Protocol II complements the definition by differentiating armed conflicts to internal disturbances. According to Article 1, armed conflicts must reach a minimum level of intensity (either being from collective character or involving governmental military forces, for instance), and non-governmental groups must have organized armed forces, with command structure and military capacity [24].

Combining both Articles, non-international armed conflicts can be defined by international humanitarian law as the following:

Protracted armed confrontations occurring between governmental armed forces and the forces of one or more armed groups, or between such groups arising on the territory of a State [party to the Geneva Conventions]. The armed confrontation must reach a minimum level of intensity and the parties involved in the conflict must show a minimum of organization [23,24].

*Understanding Narco-Trafficking as Armed Conflict*

It is estimated that in Rio de Janeiro, 56,600 criminals hold high calibre weapons, surpassing the total number held by military police officers (44,000), of which only 22,000 work in combating violence [25]. In a Brazilian armed forces operation against drug trafficking, 58,000 military troops, along with 90 military police officers, were sent to Niteroi, Rio de Janeiro to siege, stabilize and clear the area, leading to aerial restriction. Some of the equipment used included armoured car and helicopters [26].

The PCC, considered the biggest faction in Brazil, has around 30,000 members, dominating the prison system and drug market in eight states. It fights for control in 13 plus the federal district and is disadvantaged in five other states. Within the 10 states with higher mortality rates, five have a "medium presence" of PCC, leading to confrontations with other factions. It is estimated that the faction "baptized" 18,000 new members, 3000 being from the state of São Paulo and 15,000 from other states [27].

The state of Ceará is marked by the conflict between factions. Because of its important ports, Pecém and Mucuripe, the state is important for narcotrafficking logistics as an exporting centre of cocaine to Europe and Africa. With its capital holding the title of second city with the highest number of violent deaths, 77.3 victims for every 100,000 habitants, it is the scene of tension for the country's biggest factions along with local ones (GDE—Guardiões do Estado) disputing hegemony. The PCC and GDE were responsible for a slaughter of 14 people [27].

The reality in Ceará is similar to many other states in the country. In the year of 2017, the total number of violent deaths in the country was of 63,880, equivalent to 175 deaths per day or seven deaths per hour. Compared to the year of 2016, there was an increase of 2.9%. There was also an increase of deaths resulting from police actions, with 5100 in 2017, 20% higher than the year of 2016. Rio de Janeiro has the highest rate with 6.7 deaths per 100,000, followed by São Paulo, with a rate of 2.1. This increase is directly related to openly declared war between PCC and CV, fomented by the desire to increase dominated territories [1].

Despite the rivalry, the PCC influenced the other factions in the way they organize themselves and how they act. The "prison gang" style, in which the drug market is articulated from the inside of prison institutes, with statutes, notices, and even its vocabulary is diffused nationally [28].

The argument that Brazilian drug factions cannot be classified as armed conflict starts with the premise that first, the state only plays a secondary role, and the factions are not at war

in the favelas they control, but rather provide security, entertainment, social support and stimulate the local economy. Additionally, the repressive public security against these communities made them move away from police jurisdiction for all but most serious crimes [11].

Second is the mistaken notion that the drug factions in Brazil are primarily economically driven [29]. It is important to keep in mind the contexts in which factions were created in the first place, heavily tied to the notion of conflict with the state. Both the PCC and the CV, which are precursors to the uprising of other factions active in the country, arose in the prison system in which inhumane treatments were and are common towards the inmates [14]. Both criminal factions had the goal to fight the ill treatment of the prisoners, explicitly seen as both factions emerged out of prison rebellions. Rather than being economically driven, the profits of these illicit markets are understood as a method to finance their ideological goals [14].

However, being economically driven does not constitute enough reason to exclude the Brazilian situation from the armed conflict definition, as stated in International Law. There has been an increase in the debate of the economic motivations as being one of the leading causes of war, as recent wars have been heavily based on policies of aggression and conquest that would economically benefit parties [30]. The idea that wars are primarily politically motivated fails to acknowledge that wars can occur without trappings of political systems or political entities. Wars are not necessarily spontaneous acts of violence. They are organized; involve groups with different identities (regardless of it being family relationships, ethnic, religious, national, ideological, et cetera); and have a purpose other than violence, such as obtaining resources, eliminating competition for resources, or retaliating for previous attacks [31]. The conflicts between all the factions in Brazil fit this description.

## 5. The Children of Narco-Trafficking

Drug trafficking in Brazil established itself in low-income communities where it provides an important labour alternative for local youth. In a reality where the only labour options available are those at the margins of society, extreme poverty tempts children to become child soldiers and drug traffickers. Being part of a faction is an option to escape the vicious cycle of ill-paid jobs, harassment, exhausting working hours, and unsanitary work environments.

According to the last Census data of 2010, there were 11.4 million Brazilians living in favelas or in analogous conditions [32], implying a high exposure of these people to the world of drug trafficking. Most of the population living in these conditions were women. The Rocinha favela, in Rio de Janeiro, is the biggest one, with 69.161 total population and 23,352 households, with the average of 3 people per household. The northeast region of the country has 28.7% of the total number of habitants in favelas, followed by the north region with 14.4%, south with 5.3% and centre-west 1.8% [32].

The 2010 Census also provided information regarding the total number of children living in favelas or analogous situations named "abnormal agglomerations." In Brazil, there were 3,936,869 children living in favelas. See Table 1. In a reality in which the presence of factions in favelas is inevitable, this number comprises the number of potential children that could join these factions, some of whom are already members.

**Table 1.** Population living in favela households in each Regional Division—2010 [33].

| Regions | Total Population | Total Population Living in Favelas | Children and Adolescents from 0 to 17 y.o. in Favelas |
|---|---|---|---|
| North | 15,864,454 | 1,849,604 | 676,920 |
| Northeast | 53,081,950 | 3,198,061 | 1,054,615 |
| Southeast | 80,364,410 | 5,580,869 | 1,913,418 |
| South | 27,386,891 | 590,500 | 214,235 |
| Center-West | 14,058,094 | 206,610 | 77,672 |
| **Brazil** | **190,755,799** | **11,425,644** | **3,936,869** |

### 5.1. Why Children Join the Factions

Children were involved in drug markets since the emergence of factions in the 1980′s. Their roles varied but were always related to the selling of drugs within communities. In the past, their labour was paid in gifts rather than in money. The emergence of cocaine in the drug market changed the labour dynamics for children in factions, who then started to have a fixed salary depending on their roles. The highly profitable drug market led to a diffusion of factions in favelas across Brazil, contributing to an increase in disputes among different factions seeking control. With the increase in incarceration and death of those involved in the armed conflicts, children started to substitute for older members in roles that previously were mainly for adults [20]. Additionally, because children often lower suspicion and cannot be admitted into socio-educative institutions until they are 11 years old, recruitment of young children for factions, especially as soldiers, becomes attractive [26].

Social, cultural, and economic factors that lead children to join factions is tied to a lack of sufficient education and their desire to financially help their families. In a reality where formal labour is rare and inaccessible for those with low education levels, informal labour and drug trafficking are often the only options [34]. This leads to a vicious cycle in which children in poor living conditions abandon school to seek jobs and help their families. In 2019, 1.5 million Brazilian children were not enrolled in school, most of them being poor, Black, indigenous and *quilombolas* [35]. Drug trafficking presents itself as a lucrative activity for children with low education levels [34] and where jobs available for adolescents are scarce in the formal market [20].

A second reason that youth living in low-income communities may join factions is the possibility for social ascension and ability to purchase material goods. Marketing that has targeted the younger population since the 1970s intensifies the desire to own products such as a trending brand, despite financial struggle [20]. The possibility to socially ascend and obtain autonomy also fuels youths' desires to join factions [34].

There is also a subculture in favelas that praises faction members. The normalization of faction presence in favelas turned drug traffickers into idols and role models to the younger population, as people that refuse to struggle due to poverty [20]. Trafficking then can be understood as a communal practice, enabling the youth seeking recognition to be part of the "cool kids" who participate in parties and an ostentatious lifestyle [34].

The family background of children and adolescents involved in drug trafficking can also be classified a reason why they join. The increased number of single parent families, especially those with single mothers, is seen as a fundamental reason that makes children turn to trafficking [20]. The increase of family members in prison institutes or killed or in homeless conditions are other contributing factors. Non- traditional family configurations, closer to a "communal" lifestyle and suffering from domestic violence, are common themes, in which poor social conditions and everyday life marked by State actions such as relatives in prison produce "broken families" [34].

### 5.2. Children's Roles in Factions

A common factor in different factions is the vertical social ascension process in which members and employees can be promoted. To receive a promotion, those affiliated with factions are constantly evaluated to see their "readiness" to ascend in the hierarchy. The children considered ready must show a number of skills similar to any military organization, such as being trustworthy, being able to follow and complete orders, knowing how to use weapons, being able to kill, bravery, and being able to hold information [20].

The Table 2 lists the different positions children and adolescents may have in factions. Although there are other positions higher in the hierarchical chain, these are the most common ones for younger members.

Table 2. Different Roles Children and Adolescents Have in Factions [20].

| Role | Description/Function | Salary |
|---|---|---|
| Olheiro/fogueteiro (watcher) | Alert the faction of police or other factions' invasions. 'Olheiros' are put in strategical locations to guard and watch those who enter and exit the favela. They often use radios, fireworks, or both to notify the others. After notifying, they are expected to either help defend the territory or hide. | R$20 to R$50 per day (4 to 10 US dollars) |
| Vapor ('steam', seller) | Sell the drugs. A favela can have up to 15 selling points and each has a couple of 'vapores' responsible for the selling. The loads are distributed by the selling managers, and they take shifts, either alone or in groups. | R$3, R$5, or R$ 10 per bag sold (0.6 to 2 US dollars) |
| Gerente de boca (selling manager) | Responsible for supervising the drug sells, selecting "vapores" and "olheiros", loading distribution, number of employees, volume of selling, money collection, salary payment and sometimes selling drugs as well. | Not available |
| Soldado (soldier) | Responsible for maintaining order in the community, protection of members and selling points. They are always armed and can also get involved in criminal activities outside the favelas, such as stealing cars. The majority are between 15 to 17 years old. | R$1500 to R$2500 monthly (300 to 500 US dollars) |
| Fiel (loyal) | Personal bodyguard of important people in the faction. They always carry weapons and are only assigned this position when there is a strong trust bond. | Not available |

Those who decide to join drug factions do so voluntarily, even though this notion is controversial, once the life conditions make drug trafficking seem as the best (and at times the only) option available. The average age for children to join factions is 13 years old, but the process in which they begin can start when they are 8. Even though children do not initially carry weapons, it is not uncommon to see a 13-year-old with a weapon [20].

The members are expected to show constant readiness against attacks by police officials and rival factions. Weapons are given to those considered competent to defend the faction's territory, even when they are in lower rank positions such as *olheiro* and *vapor*. Because they serve as security and faction soldiers, children are actively involved in gun shootings and violence, leading to an increase in number of deaths by gun violence among minors [20].

## 6. Breaking down the 'Child Soldier' Reality

The Paris Principles on the Involvement of Children in Armed Conflicts (2007) defines a child soldier as a "child associated with an armed force or armed group" below the age of 18, boy or girl, who was or still is currently recruited or used by an armed force or armed group in any capacity: cooks, porters, spies or sexual purposes" [7].

International humanitarian law prohibits the usage of children in any form of hostilities, either international or non-international armed conflicts. Article 77 of the Additional Protocol I bans the recruitment of children below the age of 15, while Article 4 of Additional Protocol II prohibits the recruitment of volunteer enlistment. Although for international armed conflicts the prohibitions cover direct participation in hostilities, for non-international armed conflicts any form of participation in hostilities is prohibited [23,24].

Article 38(3) of the Convention on the Rights of the Child bans the recruitment of children below the age of 15 as well as voluntary enlistment [36]. The Optional Protocol to the Convention on the Rights of the Child establishes a difference in age-limits for recruitment and use of children in hostilities between States and non-State armed groups: the age-limit for direct participation in hostilities and compulsory recruitment is 18 for States, but they can accept volunteer enlistment of persons between the ages of 15 and 18. However, armed groups are bound by a strict prohibition of both voluntary and compulsory recruitment of under-18 [24]. Under the International Criminal Court Statute, Article 8(2)(b)(xxvi) and

(e)(vii), conscripting or enlisting children into armed forces or armed groups constitutes a war crime [24].

*How the Reality of Children Involved in Narco-Trafficking Can Be Classified as Child Soldiers*

Children's involvement with armed groups is often due to their vulnerable situations, a result of economic and social pressures. Poverty, discrimination, and lack of education make them an easy target for armed groups. Joining such armed groups can seem an easy solution for children living in harsh family environments and orphans [37]. Being poor, displaced, separated from relatives, and living in combat zones makes children particularly vulnerable to armed groups [38].

In some cases, children are forcibly recruited by being abducted, threatened, or coerced to join, though money and drugs are also an attraction [38]. In most cases, children voluntarily join such groups. In cases of forced recruitment, the recruiters target locations where children are most vulnerable and gathered in large numbers, such as schools, orphanages, refugee camps, stadiums, and churches [37]. Because they are more easily manipulated, do not require large amounts of food, do not have a developed sense of danger [38], and are more obedient, they tend to be a preferred target by armed groups [37].

The participation of children in armed conflict does not always entail their involvement in the hostilities. Boys and girls as young as 8 to 9 years old can also occupy support functions, such as cook, spies, messengers and sex slaves. Beyond being combatants, they can be used in acts of terror, including suicide bombing. Regardless of their position in armed conflicts, these children are exposed to high levels of violence, either as witnesses, direct victims, or forced participants [39,40]. Many child combatants in the Central African Republic were forced to execute their own parents as a form of initiation that hardens the children to commit brutal acts and break existing bonds with their communities, making their reintroduction back into society harder [38].

## 7. Protecting Children Involved in Narco-Trafficking in Accordance with International Humanitarian Law

Between the years of 1980, with the intensification of narco-trafficking and rise of drug factions in Brazil, and 2001, there was a 1340% increase in imprisonment of minors in juvenile prisons related to drug trafficking: from 110 in 1980 to 1584 in 2001 [20]. Despite the country's extensive protectionist legislation on children and adolescents, it fails to protect them. According to UNICEF, it is necessary to implement public policies to overcome geographic, social, and ethnic inequalities in the country [35].

### 7.1. The Brazilian Socio-Educative System

The Sistema Nacional de Atendimento Socioeducativo (SINASE—national system of socio-educative service), which regulates the execution of socio-educative measures to adolescents who commit infractions, bases itself on two principles: intersectionality and institutional incompleteness. The guarantee of rights for children and adolescents are, therefore, the responsibility of public policies in which the plans of the socio-educative system are articulated to serve different social operational segments: education, health, social assistance, culture, and labour and sports training [34].

Although it is expected that the rights of all children be protected by various international agreements, the institutional intersectionality which characterizes the Brazilian socio-educative system poses an obstacle to the effectiveness of the intersectoral social measures. The lack of institutional operational instruments between the different social operational segments (education, health, assistance, et cetera) combined with highly bureaucratic social agents, leads to differing understandings and proposed measures, different categorization processes and judgments, and different concepts of deviance and normality being applied by these operational segments [34], which compromises the effective and cohesive implementation of social-educative measures to children. Beyond the formal institutional arrangements, the social-educative systems are also characterized by the

subjectivity of social conceptions of rights and services by social agents and professionals, perpetuating the vulnerability of adolescents in the legal system. Few professionals and institutions implement a holistic approach (inscribed in Brazilian constitution) of social-educative measures: police and educational systems often pose children involved in drug trafficking as criminals, rather than minor offenders (as they are described in the Constitution); whereas healthcare providers often focus on substance abuse [34].

For the children and adolescents apprehended in juvenile detention centres, the socio-educative centres are seen as prisons, with the measures meant to be for pedagogical purposes seen as sentences, enforcing a negative view of institutional education and the formal labour market and a positive view of faction activities. This perception, combined with the social stigma of completing socio-educative measures, leads children to lose old social ties and have difficulty reinserting themselves back into society [41].

Systems of oppression and submission within the socio-educative centres are not uncommon, even when they are prohibited by law. The use of uniforms, line formations with children's hands behind their backs, daily body searches, rigid schedules, sanctions, the requirement to keep their heads down, among other problematic examples, evidences the negative regiment of socio-educative institutions. Some of the punishments implemented by officials include warnings, aggressions and *tranca*: adolescents are locked in their rooms for long times during the days for weeks or even months. Even though sectors linked to human rights have been fighting against the usage of *tranca* as a punishment, the practice continues. Official documents state that socio-educative measures such as training courses and formal classes are not interrupted while the adolescent is in *tranca*; however, this does not correspond to the reality [41].

The everyday-life in socio-educative centres is different for boys and girls. Among boys, PCC-like organization is the biggest trend. Rebellions and debates against the punishment measures are recurrent as ways to destabilize the institutional order, and often lead to repressions by officials such as beating sessions. Among girls, the daily life at socio-educative centres is marked by overcrowded centres; daily aggressions by officials; job trainings focusing on domestic labour and reintroduction to a household life; punishment and silence over homosexual relations; medication and transference to psychiatric centres; and free access of male officials to institutions, leading to abuses, even though it is prohibited by law [41].

### 7.2. Protections under International Humanitarian Law

International humanitarian law protects children as civilians: as members of the civilian population, they receive additional protection due to their vulnerability against the effects of hostilities, any form of indecent assault or any form of danger due to the circumstances [23,24]. The provisions contained in the Geneva Conventions and their additional protocols include protections from the effects of hostilities such as evacuation; sanitary zones; special aid and care (food, clothing, and food); protection of personal status, family, and community ties; and cultural environment and education [42].

International human rights also prevent the participation of children in hostilities; however, they still benefit from rights even if participating in hostilities. In cases where they are captured, they receive preferential treatment; and if they are members of armed forces, they benefit from combatant and prisoner-of-war status [23,24].

The 'Optional Protocol to the Convention on the Rights of the Child on the involvement of children in armed conflict' beyond prohibiting the recruitment of children under the age of 18, the conscription of soldiers below the age of 18, and recruitment of children bellow the age of 18 by armed groups also establishes that:

States should take all possible measures to prevent such recruitment—including legislation to prohibit and criminalize the recruitment of children under 18 and involve them in hostilities.

States will demobilize anyone under 18 conscripted or used in hostilities and will provide physical, psychological recovery services and help their social reintegration [41].

Brazil signed the "Optional Protocol to the Convention on the Rights of the Child on the involvement of children in armed conflict" on 6 September 2000; and ratified it on 27 January 2004 [39].

### 7.3. International Efforts and Contributions

International organizations including the United Nations have been active in battling the recruitment of children in armed conflicts. Along with using protocols in preventing children being active in hostilities, it has also been focusing on other measures. Ensuring peace processes, such as peace talks and ceasefire negotiations, has demonstrated an effective method to start implementing protections. Multiple different occasions have shown that parties to conflict can agree on the protection of children, regardless of whether they agree on everything else. A second measure is the promotion of campaigns, such as the 'Children, Not Soldiers' and 'Act to Protect Children Affected by Conflict'. Children not Soldiers, launched by the Special Representative and UNICEF, engaged with different governments in creating action plans with the United Nations in preventing usage of children in conflicts. The campaign had significant improvements and reduction in verified cases of recruitment and use of children by national security forces. Likewise, the Act to Protect aims to invigorate efforts and raise awareness about the core mandate and six great violations [43]. It will last until 2022 and has been launched in in South Sudan, Somalia, Mali, the Central African Republic, New York, Brussels, and Bangkok.

Another effort has been the engagement with non-State armed groups and parties to conflict. Most cases of violations against children in armed conflicts have systematically been promoted by non-State armed groups. In a scenario where dialogue can be difficult, the group's nature, operational environment, aspirations, and objectives need to be considered. When governments facilitate the dialogue between the United Nations and non-State armed groups, dialogue and common goals have been easier to be achieved. Parties to conflict to be removed from the Secretary-General's report on children and armed conflict must engage with action plans designed to end and prevent violations, which should be successfully implemented [43].

Action plans have been shown to be an effective measure. These are written and signed commitments between United Nations and parties listed as having committee grave violations against children designed to address specific contextual parties' situations. Commitments outline concrete and time-bound steps that lead to compliance to International Law and more a protected future for children [43]. Some of their efforts include the following:

- Criminalize the recruitment and use of children by armed forces and issue a military order to stop and prevent child recruitment
- Investigate and prosecute those who recruit and use children
- Appoint child protection specialists in security forces
- Release all children identified in the ranks of security forces
- Provide regular, unimpeded access to military camps and bases so child protection actors can verify that no children are in the ranks
- Provide release and reintegration programmes for children
- Strengthen birth registration systems and integrate age-verification mechanisms in recruitment procedures
- Implement national campaigns to raise awareness and to prevent the recruitment of children" [43].

Since the Children and Armed Conflict mandate, 33 action plans have been signed, 12 by Government forces and 21 by non-State armed groups. Twelve have been successfully complied with and consequently de-listed [43].

In Brazil, rights violations, aggression and homicide of poor adolescents and children is banal and normalized. In a society which lacks opportunities for those in vulnerable positions, poor communities turn to activities at the margins of society, while such activities are depreciated. Violence becomes the only way possible to claim visibility and achieve

social mobility. By providing protections the goal is to humanize the children and adolescents involved in drug factions, opening new possibilities without the stigmatization and demonization that condemns them to turn to a world of violence [44].

## 8. Conclusions

The factions emerged in Brazil out of a context of political instability and high demand for cocaine in the world. The inhumane treatments of inmates in prisons fuelled the creation of the first Brazilian faction, the Comando Vermelho, which had strong ties to guerrilla group ideology, organization, and methodology. The Primeiro Comando da Capital, the most influential faction in Brazil, basing itself on the principles of the Comando Vermelho to create its highly organized structure, works almost as an enterprise. Even though conflicts between the numerous factions have always been common, the year 2016 marked a new era of conflict, with the declared war between Comando Vermelho and Primeiro Comando da Capital. In an attempt to control the factions, criminalization and repression by the police only fuelled the existence of factions in the country.

In order to be classified as an armed group, according to international humanitarian law, a group must have confrontations, be on a territory of a State party, have a minimum level of intensity and have a minimum level of organization. Because the factions are constantly in conflict with each other and with the state, have control over different areas and at times over entire states, are the principal factor to the high number of violent deaths in Brazil, and are extremely organized around a hierarchy, they can be understood as armed groups. The classification of drug factions as armed groups allows the children involved in their activities to be classified as child soldiers and benefit from the protections available through international humanitarian law [43].

With the increase of criminalization and conflict between criminal factions and police throughout the years, more members were getting killed or arrested. This led to a shift in demographics in factions, with a higher number of children involved in drug-trafficking. The children and adolescents involved in drug trafficking are poor, vulnerable and at the margins of society, making drug trafficking an appealing solution to escape their living conditions. Other factors, including culture and family conditions, contribute to their engagement. The children start their socialization process with a faction at an early age; and as they show maturity and capacity to execute orders, they get promoted and officialised within the faction. Even though most of the faction leaders agree that factions are no place for children, they do not oppose their presence.

The condition of children in the drug factions in Brazil is analogous to those classified as child soldiers in international humanitarian law [7]. These are children, boys and girls, below the age of 18, who are currently recruited or used by an armed force or group, executing different functions beyond being soldiers. These are also vulnerable children, living in regions affected by war, with poverty and lack of access to resources, who either forcibly or voluntarily join the armed groups and often find their reintroduction back into society hard due to stigma. Their experience is marked by violence, either as witnesses or as victims.

Under human rights, child soldiers are given protection. Brazil has ratified the convention on Children and Armed Conflict [43]. However, the legal efforts being made have been inefficient. First, Brazil's intersectional characteristics make the implementation of socio-educative measures subjective to those who determine the sentence. Secondly, the ill treatment of adolescents in socio-educative centres, marked by repression, violence, abuse, and perpetuation of misogynist and sexist views contribute to youths' negative views towards the institution, formal labour, and education and their positive view of factions. Some of the cases in which there were interventions with respect to international organizations have been shown to have positive outcomes, in particular those in which action plans were implemented [43]. The United Nations has successfully created action plans with both governments and armed groups, half of which have been successfully accomplished.

Brazil could be the next successful story. The aim of this paper is to spark a debate on the issue and bring attention the situation of the children. We argue that with an international collective effort, there might be an increase in positive life conditions for those both in socio-educative centres and directly involved in the conflict in Brazil. Last and foremost, the research aims to change the demonization of Brazilian children involved in narco-trafficking and to promote their rights to protections offered to child soldiers under international laws.

**Author Contributions:** Conceptualization, V.B.F.D.; methodology, J.L.M.; validation, V.B.F.D. and J.L.M.; formal analysis, V.B.F.D.; resources, V.B.F.D. and J.L.M.; data curation, V.B.F.D. and J.L.M.; writing—original draft preparation, V.B.F.D.; writing—review and editing, J.L.M.; supervision, J.L.M.; project administration, J.L.M. All authors have read and agreed to the published version of the manuscript.

**Funding:** This research received no external funding.

**Informed Consent Statement:** This review involved no human subjects.

**Data Availability Statement:** Restrictions apply to the availability of these data. Some of the data was obtained from databases subscribed to by the University of Florida and are available from the authors with the permission of databases mentioned in the Methods section. Some of the information, such as news articles and international documents, are freely available through a Google search.

**Conflicts of Interest:** The authors declare no conflict of interest.

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
