# Peer review of "Children and Drug Trafficking in Brazil: Can International Humanitarian Law Provide Protections for Children Involved in Drug Trafficking?"

_societies, doi:10.3390/soc12060183_

Round 1

Reviewer 1 Report

Even though this paper covers an important topic, this cannot be taken as research (as framed by the authors in lines 93 to 96) because the materials and methods section lacks basic information for a review article such as: search dates, inclusion/exclusion criteria of studies, screening and data extraction processes, etc. As such, it is difficult to know where the results are coming from (no description of included material either). For example, it is unclear how many of the articles discussed are a) empirical b) theoretical c) grey literature. And from the empirical studies, how many (if any) included children/young people as research participants/subjects.  I suggest that much more transparency is needed for this article to be considered for publication. 

Author Response

Thank you for your review. The first author added a methods section on p 5-6 that explains the determination of the materials chosen for this research and the time period. Because no case studies are explicitly discussed, I (2nd author) could not determine an appropriate place to include numbers of children interviewed in the literature. The first author is currently in Brazil and though she responded to your initial request, I have not been able to get back in touch with her in a timely manner. We hope that our addition will sufficiently address your concerns.

Reviewer 2 Report

Thank you for referring this excellent article to me for review.  I found it fascinating and learned a lot about the situation facing children caught up in Brazil's drug trafficking cartels.  I like the argument made about the relevance and use of humanitarian law very much.  In fact, I would like to see this article read widely and used as a real conversation starter.  The expression, referencing and command of relevant literature seems fine to me.  My only suggestion is that the discussion at the end of the piece is a little short.  I wonder whether it might be worth reflecting on the impact of the Bolsonaro government.  The research was conducted during COVID-19.  Were children more susceptible to harm during this period?  These are suggestions only: the article is fine for publication as it is.  It is a really original piece with the potential to reach a large audience.  The author/s are to be congratulated on a fine piece of work.

Author Response

Thank you for your kind review. I agree that it could be helpful to learn about how the Bolsonaro government affected the situation. I tried four data bases and found nothing, unfortunately. With respect to COVID, I found "Brazil: Hate and intolerance in times of pandemic in a mixed-race country," by Claudia Morelli Gadotti and Vera Lucia Colson in the Journal of Analytical Psychology (2021), but it was not helpful to the paper. The first author is currently in Brazil and I have not been able to hear back from her in a timely fashion. We appreciate that you feel the manuscript is publishable even without these additions.

Reviewer 3 Report

The article is well-written and tackles a serious issue within the Brazilian society. It would be great if the author could present the solutions proposed by the different ruling political parties and their outcome in the framework of the ongoing presidential elections. Additionally, a comparative analysis with case studies in countries where children-soldiers are a reality during wartime would have improved the paper. Overall the paper is exemplary and should be published!!!

Author Response

Thank you for your kind review! Your comment about the effects of the ruling parties is similar to one we received from another reviewer. I looked at four academic databases and came up with nothing on this issue, most unfortunately. The first author is currently in Brazil, and I have not been able to get a response from her in a timely manner. 

The idea of comparative case studies is an interesting and fruitful one. I believe that we would have to re-structure the whole paper and research questions to write a comparative case study. I conducted research in Uganda  with former child soldiers for five years after the LRA War, but these cases are so complex and different from how Brazil views the children dealing drugs that it would be difficult to include without adding significant length and redirection to the paper. As such, we chose to refer primarily to international reports, treaties, and conventions.

Round 2

Reviewer 1 Report

Thank you for clarifying the sources of your data, this makes your article much more transparent and shows it is indeed emprirical research.

You cover an important topic and contribute to an urgent debate to find ways to protect children.

Author Response

Thank you for your helpful reviews!